# Dysfunction of the Lenticular Nucleus Is Associated with Dystonia in Wilson’s Disease

**DOI:** 10.3390/brainsci13010007

**Published:** 2022-12-20

**Authors:** Yulong Yang, Taohua Wei, Wenming Yang, Sheng Hu, Hailin Jiang, Wei Dong, Wenjie Hao, Yue Yang, Nannan Qian, Yufeng Ding

**Affiliations:** 1Graduate School, Anhui University of Traditional Chinese Medicine, Hefei 230038, China; 2Neurology Department, The First Affiliated Hospital of Anhui University of Traditional Chinese Medicine, Hefei 230031, China; 3Centers for Biomedical Engineering, University of Science and Technology of China, Hefei 230026, China; 4School of Medical Information Engineering, Anhui University of Chinese Medicine, Hefei 230038, China

**Keywords:** Wilson’s disease, lenticular nucleus, dystonia, functional connectivity, unified Wilson’s disease rating scale, muscle tension

## Abstract

Dysfunction of the lenticular nucleus is thought to contribute to neurological symptoms in Wilson’s disease (WD). However, very little is known about whether and how the lenticular nucleus influences dystonia by interacting with the cerebral cortex and cerebellum. To solve this problem, we recruited 37 WD patients (20 men; age, 23.95 ± 6.95 years; age range, 12–37 years) and 37 age- and sex-matched healthy controls (HCs) (25 men; age, 25.19 ± 1.88 years; age range, 20–30 years), and each subject underwent resting-state functional magnetic resonance imaging (RS-fMRI) scans. The muscle biomechanical parameters and Unified Wilson Disease Rating Scale (UWDRS) were used to evaluate the level of dystonia and clinical representations, respectively. The lenticular nucleus, including the putamen and globus pallidus, was divided into 12 subregions according to dorsal, ventral, anterior and posterior localization and seed-based functional connectivity (FC) was calculated for each subregion. The relationships between FC changes in the lenticular nucleus with muscle tension levels and clinical representations were further investigated by correlation analysis. Dystonia was diagnosed by comparing all WD muscle biomechanical parameters with healthy controls (HCs). Compared with HCs, FC decreased from all subregions in the putamen except the right ventral posterior part to the middle cingulate cortex (MCC) and decreased FC of all subregions in the putamen except the left ventral anterior part to the cerebellum was observed in patients with WD. Patients with WD also showed decreased FC of the left globus pallidus primarily distributed in the MCC and cerebellum and illustrated decreased FC from the right globus pallidus to the cerebellum. FC from the putamen to the MCC was significantly correlated with psychiatric symptoms. FC from the putamen to the cerebellum was significantly correlated with muscle tension and neurological symptoms. Additionally, the FC from the globus pallidus to the cerebellum was also associated with muscle tension. Together, these findings highlight that lenticular nucleus–cerebellum circuits may serve as neural biomarkers of dystonia and provide implications for the neural mechanisms underlying dystonia in WD.

## 1. Introduction

Wilson’s disease (WD) is a rare inherited disorder of copper metabolism, resulting in copper accumulation in many organs, particularly in the liver and brain [1]. Neurological symptoms are one of the most frequent clinical symptoms in WD, including tremors, dystonia and parkinsonism [2]. The basal ganglia are the most severely affected brain regions in patients with WD [3,4]. The lenticular nucleus, as an important part of the basal ganglia, plays an essential role in motor control, emotion and motivation [5,6,7]. Evidence suggests that the lenticular nucleus is vulnerable to degeneration in Parkinson’s and focal dystonia disease, which leads to dysfunction of the cortico-basal ganglia–cerebellum circuits that contribute to dystonia in those patients [8,9]. Neuroimaging studies have reported abnormalities in the basal ganglia in WD, particularly caudate and putamen nucleus shrinkage [3,10]. However, no direct neuroimaging evidence exists to characterize how dysfunction of the lenticular nucleus affects dystonia in WD.

Currently, neuroimaging serves as a noninvasive tool to evaluate brain abnormalities in WD [3,10,11]. Recent WD-related brain structural studies revealed gray volume atrophy and local shape abnormalities in the putamen, globus pallidus and caudate, and these abnormalities affect neurological symptoms in WD [11,12]. Resting-state functional magnetic resonance imaging (RS-fMRI) studies also reported that dysfunction in the lenticular nucleus can serve as a neural biomarker to classify neurological WD patients and healthy controls [13] and demonstrated that abnormal functional connectivity (FC) from the lenticular nucleus to the cerebral cortex and cerebellum characterizes the imaging phenotype of neurological symptoms in WD [14]. Dystonia is mainly caused by the impairment of muscle tension, leading to abnormal facial expressions and limb movement disorders in WD [1,15]. Although the imaging phenotype of neurological symptoms is well investigated, dystonia, as one of the most frequent neurological symptoms, has not been well studied, and its clinical-related imaging features have not been revealed in WD.

Here, we hypothesized that the dysfunction in the pathways from the lenticular nucleus to the cerebral cortex and cerebellum play crucial roles in dystonia of WD. To test this hypothesis, we employed RS-fMRI maps to calculate seed-based FC for the lenticular nucleus, and the relationships between FC changes and dystonia and clinical symptoms were further evaluated to characterize neural imaging features of dystonia in WD. This is expected to provide new insights into the mechanism of dystonia and instruction for further research on dystonia in WD.

## 2. Materials and Methods

### 2.1. Participants

Thirty-seven native Chinese-speaking WD patients with extremities dystonia and thirty-seven age- and sex-matched HCs were recruited from the First Affiliated Hospital of Anhui University of Traditional Chinese Medicine (AUTCM) between April 2021 and April 2022. Patient inclusion criteria were as follows: met the diagnostic guidelines for WD established by the European Association for Liver Research in 2012 [16]; 12–40 years old; normal communication, audio-visual reading and writing abilities; or diagnosed with multisegmental dystonia, mainly manifested in the extremities, by the relevant dystonic standard [17,18,19]. The exclusion criteria were as follows: obvious head shaking, which affects the magnetic resonance data acquisition; a history of craniocerebral surgery or metal implants in the body; the use of drugs that can affect muscle tension, such as baclofen and benzhexol, in the 2 months before the study; the disease in rapid progress; or other diseases that affect muscle tension and stiffness, such as brain injury, myositis, etc. Normal individuals with a history of problems in the central nervous system, mental health or other serious diseases were excluded from this study. All the WD patients we recruited met the inclusion and exclusion criteria for this study. The current study was approved by the Human Research Committee of the First Affiliated Hospital of AUTCM, and all subjects signed a written informed consent form before enrollment.

### 2.2. Clinical and Muscle Biomechanical Assessment

Evaluation of clinical symptoms of WD was performed by an experienced neuropsychologist. The Unified Wilson’s Disease Rating Scale (UWDRS), which has been used to measure the severity of WD, was performed, and the neurological (UWDRS-N) and psychiatric (UWDRS-P) examination scores were recorded. Detailed information on the UWDRS scale is provided in Appendix A.

As a special neurogenetic disease, dystonia of the extremities is also common in WD patients and mainly caused by muscle tension impairment, as reported by previous studies, and its impact on the quality of life and prognosis of patients was particularly prominent [20,21]. So, it is very meaningful to study WD with extremities dystonia. Previous studies have reported that the abnormal tension of the biceps brachii and medialis and lateralis gastrocnemius muscle is one of the commonly involved parts of extremities dystonia and can also reveal the degree of dystonia in upper and lower limbs, which can be indirectly reflected by measuring the muscle tension levels of biceps brachii and gastrocnemius muscle [22,23,24,25]. Furthermore, previous dystonia-related studies and our recent dystonia study of WD have shown that the digital muscle function assessment system (MyotonPRO) can reliably assess the degree of extremities dystonia by measuring the muscle tension levels of the biceps brachii and gastrocnemius [26,27,28,29,30]. Therefore, we selected the extremities to measure the biomechanical level of WD patients, and the muscle biomechanical level was assessed using the MyotonPRO by measuring the biceps brachii and medialis and lateralis gastrocnemius muscle on both sides to reflect the degree of dystonia.

The measurement contents included the F, S, D, R and C values. F is the frequency of muscle oscillation, representing muscle tension; S is the stiffness of the muscle; D is the decay rate, representing muscle elasticity; R is the release time of the muscle pressure, which is the time required for the muscle to recover from the maximum deformation to the original shape after being squeezed and deformed by an external force; and C is the Deborah number, which represents the ratio of the time it takes for the muscle to release pressure to the time it takes for the muscle to reach its maximum deformation. Each muscle was measured 5 times, and the mean value was calculated. The details of MyotonPRO are provided in Appendix A.

### 2.3. MRI Acquisition

In this study, a General Electric 3.0 Tesla (GE, 3.0T) whole-body magnetic resonance imaging device was used for scanning. During the scanning, the subject was required to fix the head with a sponge pad, wear headphones to reduce the impact of noise on the subjects, remain still in the supine position and close their eyes.

T1-weighted images were acquired by a T1-3D BRAVO sequence. The main parameters were as follows: repetition time (TR) = 8.16 ms; echo time (TE) = 3.18 ms; flip angle (FA) = 12°; matrix = 256 × 256; field of view (FOV) = 256 × 256 mm^2^; resolution = 1 × 1 mm^2^; slice thickness = 1 mm; a total of 170 slices were scanned. The resting-state fMRI images were obtained using the BOLD imaging sequence. The main scanning parameters are as follows: TR = 2000 ms; TE = 30 ms; FA = 90°; matrix = 64 × 64; FOV = 220 × 220 mm^2^; resolution = 3.4375 × 3.4375 mm^2^; slice thickness = 3 mm. 185 volumes were acquired from each individual, with 36 slices per volume.

### 2.4. Data Preprocessing

The preprocessing pipeline included the following steps: slice timing correction (to correct for temporal shifts of different slices), within-subject EPI image realignment (to estimate and spatially correct for head motions of different EPI volumes), rigid-body registration of each subject’s T1 image to the EPI mean image, and normalization of the EPI images to the Montreal Neurological Institute (MNI) standard space using the T1 image. After spatial normalization, the EPI images were resampled to 3 × 3 × 3 mm^3^, followed by a noise removal process, including a multiple regression model and bandpass filtering. Regressors of the regression model included linear trends, average white matter (WM), cerebrospinal fluid (CSF) and whole-brain (global signal) voxels, the first derivatives of WM and CSF, and Friston’s 24-parameter head motion model. After removing potentially noisy signals using averaged brain tissue time series and estimated head motion signals in the multiple regression model, the residuals were bandpass-filtered (0.01–0.08 Hz) to further suppress low-frequency drifts and physiological noises, such as breathing and heartbeat. All the processes were performed by DPABI [31] (http://rfmri.org/dpabi, accessed on 1 May 2021), a MATLAB toolbox for batch preprocessing fMRI data.

### 2.5. Regions of Interest

Subregions of the lenticular nucleus were selected as regions of interest (ROIs) based on a previous study [32]. Specifically, the lenticular nucleus was divided into 12 subregions (Figure 1). Each side of the putamen was divided into 4 subregions, including the bilateral ventral anterior (PUT-VA), dorsal anterior (PUT-DA), dorsal posterior (PUT-DP), and ventral posterior (PUT-VP) putamen, and each side of the globus pallidus was divided into 2 subregions containing the bilateral anterior (aGP) and posterior (pGP) globus pallidus. Each subregion of the lenticular nucleus was further selected as the ROI for seed-based FC analysis.

### 2.6. Functional Connectivity Analysis

FC analysis was performed using Analysis of Functional Neuroimage (AFNI, version 19.3.08, http://afni.nimh.nih.gov, accessed on 21 October 2019) software. For each participant, the Pearson correlation coefficient between the average time series of each seed and the time series of every voxel across the entire brain was calculated, and the coefficients were further converted into a z-value using Fisher r-to-z transformation to improve normality. Therefore, a seed-based FC map was acquired for each subregion from each participant.

### 2.7. Statistical Analysis

Before statistical analysis of muscle tension level, we standardized all parameters to the range of 0 to 1 by using the min-max normalization. First, a two-sample t-test was performed to compare muscle biomechanical parameters between the HCs and WD groups, and *p* < 0.05 was considered statistically significant. Statistical Product Service Solutions (SPSS) (version 26.0) software was used for this statistical analysis.

A group-level analysis was employed to determine FC differences between WD and HC. The results were corrected using 3dClustsim with *p* < 0.001 at the voxel level and *p* < 0.01 at the cluster level. Finally, the FC value in significantly different regions was further extracted to perform correlation analysis with UWDRS-N, UWDRS-P, and muscle biomechanical parameters. The group-level analysis and results correction were performed by AFNI (version 19.3.08).

## 3. Results

### 3.1. Patient Characteristics

The mean age of the thirty-seven patients was 23.95 ± 6.95 years (range 12–37), and the duration of extremities dystonia after WD was 9.59 ± 5.39 years (range 1–24). Thirty-seven patients had neurological symptoms, and twenty-six patients had neuropsychiatric symptoms. Thirty-four patients had a Kayser–Fleischer ring in the cornea. Table 1 summarizes the general and clinical characteristics of all patients.

### 3.2. Muscle Tension

The F values in the right gastrocnemius medialis and lateralis muscle were significantly different between the WD patients and HCs (*p* < 0.05, Figure 2). No other parameters showed significant differences between the WDs and HCs. This indicated that WD patients have a certain degree of muscle tension impairment. The F values were used for further correlation analysis with FC.

### 3.3. Group Comparison of FC between WD and HCs

Compared with HCs, WD patients illustrated that FC from all subregions of the putamen to the cerebellum and FC from all subregions of the putamen except the right PUT-VP to the middle cingulate cortex (MCC) were decreased (Figure 3A and Table 2). WD patients showed that the FC of the bilateral aGP and pGP decreased in the cerebellum compared with HCs (Figure 3B and Table 3). In addition, FC from the right aGP and pGP to MCC was decreased in WD patients compared with HCs (Figure 3B and Table 3). The FC maps of all the subregions in the putamen and globus pallidum overlapped in one map (Figure 3C), and the FC of the max overlapped regions, shown in black circle of Figure 3C, were extracted to perform correlation analysis with UWDRS-N, UWDRS-P and F values in WD patients.

### 3.4. Correlation between FC and UWDRS Score

The FC from the right PUT-DA (R = −0.5288, *p* = 0.0008, Figure 4A), right PUT-VP (R = −0.4249, *p* = 0.0088, Figure 4B) and right PUT-DP (R = −0.5466, *p* = 0.0005, Figure 4C) to the cerebellum was significantly negatively correlated with UWDRS-N but not significantly correlated with UWDRS-P. The FC from the right PUT-VA (R = −0.4811, *p* = 0.0026, Figure 4D), right PUT-VP (R = −0.4091, *p* = 0.0119, Figure 4E) and right PUT-DP (R = −0.4520, *p* = 0.0050, Figure 4F) to the MCC was significantly negatively correlated with UWDRS-P but not significantly correlated with UWDRS-N.

### 3.5. Correlations between FC and Muscle Tension

The FC from the left PUT-DA (R = 0.4627, *p* = 0.0039, Figure 5A), left PUT-VP (R = 0.4527, *p* = 0.0086, Figure 5B), left PUT-DP (R = 0.3772, *p* = 0.0214, Figure 5C), right PUT-VA (R = 0.5342, *p* = 0.0007, Figure 5D), right PUT-DA (R = 0.3876 *p* = 0.0178, Figure 5E) right PUT-VP (R = 0.4506, *p* = 0.0051, Figure 5F) and right pGP (R = 0.3810, *p* = 0.0200, Figure 5I) to the cerebellum were significantly positively correlated with the F value of the left ciceps brachii short head. Meanwhile, the FC from the left PUT-DA (R = 0.3793, *p* = 0.0206, Figure 5G) and right PUT-VP (R = 0.3363, *p* = 0.0418, Figure 5H) to cerebellum was also significantly positively correlated with the F value of the right biceps brachii short head (Figure 5B).

## 4. Discussion

The present study investigated relationships between lenticular dysfunction and dystonia in WD patients based on RS-fMRI data by seed-based FC and correlation analysis. We found that FC from the lenticular nucleus to the cerebellum was associated with muscle tension and UWDRS-N, and globus pallidum-cerebellum FC was also related to muscle tension. Our results indicated that dysfunction of lenticular nucleus–cerebellum pathways contributed to dystonia in WD patients. To the best of our knowledge, this is the first study to explore the FC abnormalities of the lenticular nucleus and associate these FC changes with dystonia in WD.

Dystonia is one of the most common representations of WD [33,34]. It can be focal (involves one body part, for example, one hand), segmental (involves one body segment, for example, upper extremity), multisegmental (involves multiple segments, for example, face and leg) or may even be generalized [35]. Multisegmental dystonia, manifested as limb dystonia, is also common in clinical practice and may affect other extrapyramidal symptoms [15,20,21]. In previous studies, extensive brain functional and structural impairments, particularly in basal ganglia nucleus, were found in other dystonia-related diseases [36,37]. Degenerative changes in the lenticular nucleus have also been reported in patients with WD, and their impairment is associated with neurological symptoms [11,38,39]. Previous studies have also demonstrated that dysfunction of the lenticular nucleus is associated with the disease severity of WD [40]. FC between the basal ganglia and MCC was decreased in WD patients, as found in a previous study [41]. In line with this study, our study also found that FC from all subregions of the putamen, except from the ventral posterior part to the MCC and FC from the right globus pallidus to the MCC, were decreased in WD patients. The MCC, part of the cingulate cortex, projects into the striatum and is involved in decision-making, emotion and motivation [6,42]. A previous study illustrated that FC between the stratum and cingulate cortex is associated with psychiatric symptoms [43], but no studies revealed that this pathway is involved in extrapyramidal symptoms. Naturally, we found there were no significant correlations between lenticular nucleus-MCC FC and UWDRS-N scores or muscle tension in WD patients. Of note, the significant correlations between FC from the right ventral anterior, ventral posterior and dorsal posterior putamen to the MCC and UWDRS-P were observed in WD patients. Therefore, we speculated that lenticular nucleus-MCC FC is not the main contributor to neurological symptoms in WD but may affect WD patients with psychiatric symptoms.

The cerebellum is thought to maintain body posture, balance, regulate muscle tension and coordinate voluntary movements [44]. Dysfunction of cerebellar–basal ganglia interactions leads to movement disorders, such as Parkinson’s disease and cervical dystonia [45,46]. A previous study found that dysfunction of basal ganglia and cerebellar circuits may play an important role in the occurrence of blepharospasm [47]. In addition, functional abnormality were found not only in the basal ganglia, but also in the cerebellum basal ganglia cortex circuit in focal dystonia [48]. According to other studies, lesions in the cerebellum and lenticular nucleus are associated with neurological symptoms in WD, which was also observed in previous studies [11,49]. Recent WD-related FC studies also showed that striatum-cerebellum FC is decreased in WD patients compared with healthy controls, and these FCs were associated with the severity of neurological symptoms in WD [41,50]. These studies indicated that lesions in the basal ganglia and cerebellum are associated with dystonia. Corresponding to previous studies, we showed that FC from all subregions of the lenticular nucleus to the cerebellum was decreased and that FC from the right dorsal anterior, dorsal posterior and ventral posterior putamen was negatively associated with UWDRS-N in WD patients. This finding indicates that the impairment of FC between the lenticular nucleus and cerebellum is related to the neurological symptoms of patients with WD. Of note, we further observed that FC from the putamen to the cerebellum and from the posterior globus pallidus to the cerebellum was positively correlated with muscle tension. The lenticular nucleus is composed of the globus pallidus and putamen, which play opposite roles in regulating muscle tension [7]. Globus pallidus lesions increase muscle tension, and putamen lesions decrease muscle tension [51,52]. As the main component of regulating human muscle tension, hypotonia will occur when cerebellar function is impaired [53]. Therefore, the higher the putamen and cerebellum FC values are, the more likely the patient’s muscle tension level will tend to be normal, which is consistent with our results. However, we also found that FC from the globus pallidus to the cerebellum was positively correlated with muscle tension, which indicated that when the FC between the globus pallidus and the cerebellum was abnormal, cerebellar function may play a leading role in regulating muscle tension. In summary, these findings indicated that lenticular nucleus-cerebellum pathways might be the neural mechanism of dystonia in WD.

These findings have some implications for future clinical research on WD. On the one hand, these findings may be helpful for the clinical treatment of WD patients with dystonia. Previous studies have found that transcrania magnetic stimulation (TMS) can be used to alleviate and treat some neuropsychiatric disorders and Wilson’s disease [30,54]. Other studies have also demonstrated that the dorsolateral prefrontal cortex and cerebellum can serve as targets to treat depression [55,56] and schizophrenia [57,58], respectively. Our study found that the lenticular nucleus–cerebellum pathway is related to dystonia in WD patients. Therefore, the results suggest that the cerebellum may serve as a potential target to treat WD patients with dystonia and neurological symptoms by TMS. On the other hand, the lenticular nucleus-cerebellum may serve as a neural biomarker to identify WD patients with dystonia and to predict disease prognosis by machine learning in future studies.

There are also some limitations in the present study. First, the sample size in the present study is relatively small. Although the current findings survive a rigorous threshold, the relatively small sample size may limit the generalization of these findings, and it should be expanded in future studies to make more robust results. Second, we found that the lenticular nucleus-cerebellum pathway is associated with dystonia in WD patients; however, how this pathway influences treatment in WD with dystonia was not studied in the present study. In the future, a long-term dataset should be applied to understand the role of this pathway in treating WD patients with dystonia. Third, the UWDRS-N is a comprehensive score reflecting neurological symptoms of WD, but it cannot be used to accurately classify the neurological symptoms. Therefore, different types of neurological symptoms require different scales to study the classification of WD in the future.

## 5. Conclusions

In conclusion, our study found that aberrant FC from the lenticular nucleus to the cerebellum and MCC were observed in patients with WD. The FC changes from the lenticular nucleus to the cerebellum were related to UWDRS-N and were also significantly correlated with muscle tension levels. These findings suggested that the lenticular nucleus–cerebellum pathways may serve as neural biomarkers of dystonia and provided implications for the neural mechanisms underlying dystonia in WD. In addition, this study has potential clinical application value in that the cerebellum may serve as a potential target for TMS to treat dystonia and neurological symptoms of WD.

## Figures and Tables

**Figure 1 brainsci-13-00007-f001:**
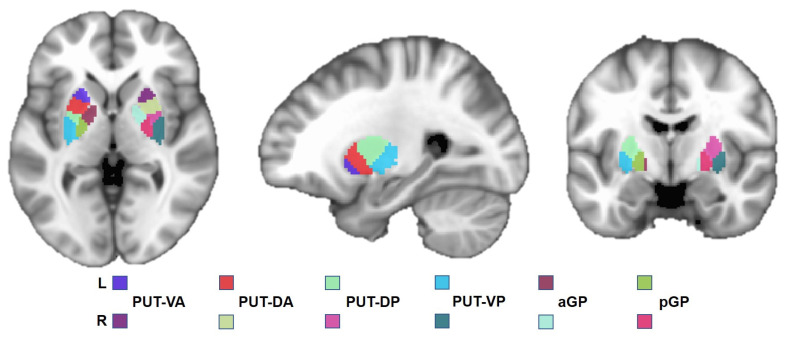
Definition of ROIs. Abbreviations: L, left; R, right; PUT-VA, ventral anterior putamen; PUT-DA, dorsal anterior putamen; PUT-DP, dorsal posterior putamen; PUT-VP, ventroposterior putamen; aGp, anterior globus pallidus; pGP, posterior globus pallidus.

**Figure 2 brainsci-13-00007-f002:**
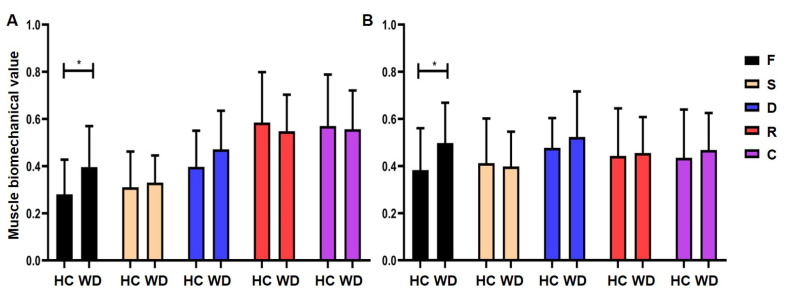
Group differences in muscle biomechanical level. (**A**) Differences between WD and HC on the right gastocnemius medialis muscle biomechanical levels. (**B**) Differences between WD and HC on the right gastrocnemius lateralis muscle biomechanical levels. Abbreviations: F, frequency of muscle oscillation; S, muscle stiffness; D, rate of decrement; R, muscle pressure release time; C, Deborah number; HC, healthy control; WD, Wilson’s disease; * *p* < 0.05.

**Figure 3 brainsci-13-00007-f003:**
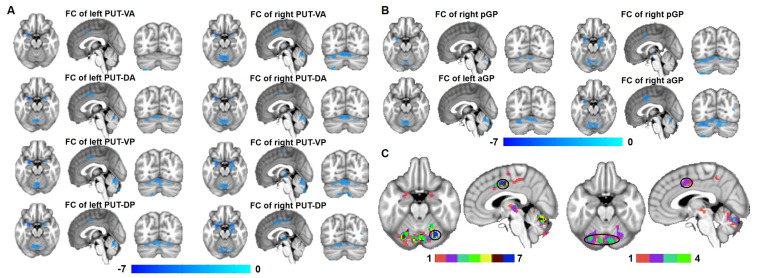
Group differences in FC of the lenticular nucleus. (**A**) FC changes in all subregions of the putamen between WDs and HCs. (**B**) FC changes in all subregions of the globus pallidus between WD patients and HCs. (**C**) FC changes in the eight subregions of the bilateral putamen and FC in the four subregions of the bilateral globus pallidus overlap in the MCC and cerebellum. Abbreviations: FC, functional connectivity; HCs, healthy controls; WD, Wilson’s disease.

**Figure 4 brainsci-13-00007-f004:**
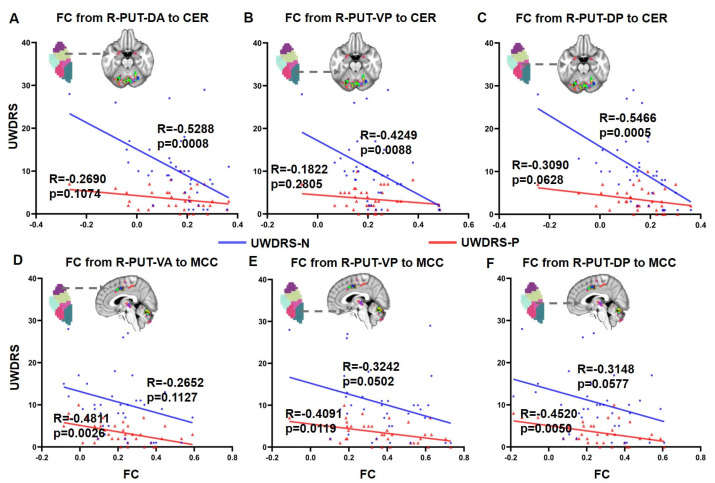
Correlation between FC from the putamen to cerebellum or middle cingulate cortex and subscales of UWDRS. (**A**) Correlation between FC from R-PUT-DA to CER and subscales of UWDRS; (**B**) Correlation between FC from R-PUT-VP to CER and subscales of UWDRS; (**C**) Correlation between FC from R-PUT-DP to CER and subscales of UWDRS; (**D**) Correlation between FC from R-PUT-VA to MCC and subscales of UWDRS; (**E**) Correlation between FC from R-PUT-VP to MCC and subscales of UWDRS; (**F**) Correlation between FC from R-PUT-DP to MCC and subscales of UWDRS; Abbreviations: FC, functional connectivity; CER, cerebellum; MCC, middle cingulate cortex; PUT, putamen; DA, dorsal anterior; VP, ventral posterior; DP, right dorsal posterior; VA, ventral anterior; L, left; R, right; UWDRS, United Wilson’s Disease Rating Scale; UWDRS-N, neurological subscale in UWDRS; UWDRS-P, psychiatric subscale in UWDRS.

**Figure 5 brainsci-13-00007-f005:**
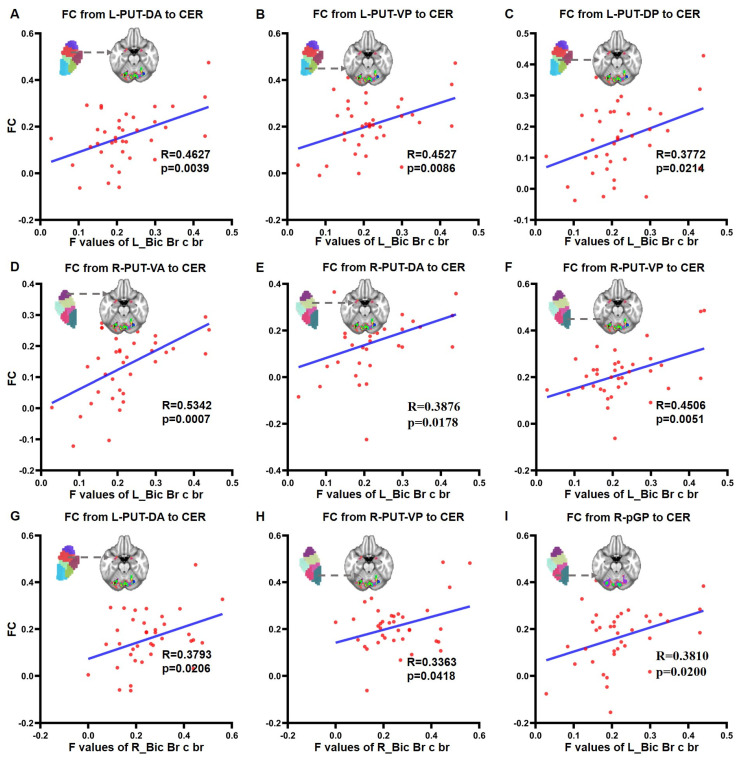
Correlation between FC from the lenticular nucleus to cerebellum and muscle tension level. (**A**) Correlation between FC from L-PUT-DA to CER and F value of L_Bic Br c br; (**B**) correlation between FC from L-PUT-VP to CER and F value of L_Bic Br c br; (**C**) correlation between FC from L-PUT-DP to CER and F value of L_Bic Br c br; (**D**) correlation between FC from R-PUT-VA to CER and F value of L_Bic Br c br; (**E**) correlation between FC from R-PUT-DA to CER and F value of L_Bic Br c br; (**F**) correlation between FC from R-PUT-VP to CER and F value of L_Bic Br c br; (**G**) correlation between FC from L-PUT-DA to CER and R_Bic Br c br; (**H**) correlation between FC from R-PUT-VP to CER and R_Bic Br c br; and (**I**) correlation between FC from R-pGP to CER and L_Bic Br c br; Abbreviations: FC, functional connectivity; CER, cerebellum; MCC, middle cingulate cortex; PUT, putamen; GP, globus pallidus; VA, ventral anterior; DA, dorsal anterior; VP, ventral posterior; DP, dorsal posterior; p, posterior; L, left; R, right; F, frequency of muscle oscillation; L_Bic Br c br, left Biceps Brachii short head; R_Bic Br c br, right Biceps Brachii short head.

**Table 1 brainsci-13-00007-t001:** General and clinical characteristics of the included WD patients.

Variables	Patient
Gender (man/woman) Symptoms (neurological/neuropsychiatric) Age (years) Handedness Duration (years) UWDRS-N UWDRS-P K-F ring (+/−) CER U-cu	20/17 37/26 12–37 (23.95 ± 6.95) 37 right-handed 1–24 (9.59 ± 5.39) 1–29 (10.49 ± 7.56) 0–10 (3.49 ± 2.55) 34/3 <0.1 g/L >100 µg/24 h

Abbreviations: UWDRS, United Wilson’s Disease Rating Scale; UWDRS-N, neurological subscale in UWDRS; UWDRS-P, psychiatric subscale in UWDRS; K-F, Kayser-Fleischer; CER, ceruloplasmin; U-cu, urine cooper; N, number.

**Table 2 brainsci-13-00007-t002:** Group FC differences in the putamen between WD patients and HCs.

ROIs	Regions	MNI Coordinate			Peak Z Value
		x	y	z	
L-PUT-VA	Left cerebellum	−26	−68	−60	−3.95
	Right MCC	4	0	42	−3.97
L-PUT-DA	Left cerebellum	0	−66	−18	−4.34
	Right MCC	4	0	44	−4.23
L-PUT-VP	Right cerebellum	30	−60	−54	−4.68
	Right MCC	2	0	44	−4.20
L-PUT-DP	Left cerebellum	−10	−64	−14	−4.65
	Right MCC	4	2	44	−4.40
R_PUT-VA	Right cerebellum	10	−64	−14	−5.04
	Right MCC	2	2	44	−4.62
R-PUT-DA	Right cerebellum	10	−66	−14	−4.78
	Right MCC	2	0	44	−4.21
R-PUT-VP	Left cerebellum	−10	−74	−48	−4.94
R-PUT-DP	Right cerebellum	28	−64	−20	−4.83
	Right MCC	2	−2	44	−4.68

Abbreviations: FC, functional connectivity; WD, Wilson’s disease; HCs, healthy controls; ROIs, regions of interest; MCC, middle cingulate cortex; PUT, putamen; VA, ventral anterior; DA, dorsal anterior; VP, ventral posterior; DP, dorsal posterior; L, left; R, right.

**Table 3 brainsci-13-00007-t003:** Group FC differences in the globus pallidum between WD patients and HCs.

ROIs	Regions	MNI Coordinate			Peak Z Value
		x	y	z	
L-pGP	R cerebellum	6	−76	−24	−4.22
L-aGP	R cerebellum	10	−76	−24	−4.99
R-pGP	Left MCC	0	8	42	−3.92
	Left cerebellum	−12	−78	−48	−4.78
R-aGP	Right MCC	2	4	42	−4.46
	Right cerebellum	−28	−72	−24	−5.14

Abbreviations: FC, functional connectivity; WD, Wilson’s disease; HCs, healthy controls; ROIs, regions of interest; GP, posterior globus pallidus; MCC, middle cingulate cortex; p, posterior; a, anterior; L, left; R, right.

## Data Availability

Not applicable.

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
