# Peer review of "Dysfunction of the Lenticular Nucleus Is Associated with Dystonia in Wilson’s Disease"

_brainsci, 2022, doi:10.3390/brainsci13010007_

Round 1
Reviewer 1 Report
The current manuscript studies the role of dysfunction of the lenticular nucleus in dystonia among Wilsons’ disease. It has been done through resting-state functional magnetic resonance imaging, and muscle biomechanical levels, and neurological and psychiatric symptoms. The study findings confirm that lenticular nucleus–cerebellum circuits may serve as neural biomarkers of dystonia and provide implications for the neural mechanisms underlying dystonia in WD. The manuscript is well written and informative. Please find my following points;
1- Line 18; ‘’ to systematically solve this problem’’, I think that this statement is not true.
2- Line 64-71; I think that is more appropriate in Methods more than introduction, you can state the hypothesis and aim of the study in the end of introduction.
3- Line 101; Please justify the rational for use biceps brachii and the medialis and lateralis gastrocnemius muscle for biomechanical analysis.
4- I strongly recommend the authors to report the study according to STROBE stamen.
5- Is the sample size based on power calculations
6- Surprisingly, there is no clinical or research recommendations based on the interesting study finding
7- Although the study is interesting and informative, the discussion is the weakest part. There is no good explanation of the results, clinical implication, or limitation
Reviewer 2 Report
The authors reported an interesting study about lenticular nucleus in patients with dystonia in Wilsons’ disease. I have some comments to the authors:
(i) Please specify whether all the WD patients had dystonia – it is not clear.
(ii) Please clarify how you made the diagnosis of dystonia. This is crucial because it’s important that you have followed the most recent recommendation. Please refer to this article also in the text:
Romano M, et al. Diagnostic and therapeutic recommendations in adult dystonia: a joint document by the Italian Society of Neurology, the Italian Academy for the Study of Parkinson's Disease and Movement Disorders, and the Italian Network on Botulinum Toxin. Neurol Sci. 2022 Oct 3. doi: 10.1007/s10072-022-06424-x. Epub ahead of print. PMID: 36190683.
(iii) The authors should better address the neurological features of the patients. Table 1 should be modified as followed: please delete the HC (the information reported in the table have been already mentioned in the text); add more info of the patients (e.g. neurological symptoms, psychiatric symptoms etc)
(iv) Why have the authors considered the score of the UWDRS-N related to all neurological symptoms? Indeed, this subscale investigated also other symptoms such as tremor etc. It is not clear how the authors have identified only dystonia (again it’s important that the authors better explain the inclusion criteria and the clinical features of the patients with WD).
(v) What is the rationale of measuring biceps brachii and the medialis and lateralis gastrocnemius muscle on both sides? The most common kind of dystonia is the cervical one, followed by blepharospasm. Noteworthy, lower limb dystonia is very rare.
(vi) The authors should specify the body distribution of dystonia according to the MDS-classification:
Albanese A, et al. Phenomenology and classification of dystonia: a consensus update. Mov Disord. 2013 Jun 15;28(7):863-73. doi: 10.1002/mds.25475. Epub 2013 May 6. PMID: 23649720; PMCID: PMC3729880.
(vii) Discussion should be rewritten following the (i) to (vi) points. The authors should also include a paragraph with the limitation of the study.
Reviewer 3 Report
1. The article type (research type) should be mentioned in the title.
2. Abstract.
a. Include the number of participants.
b. could the authors include more specific data (percentages, OR, numbers, …) in the abstract.
3. Please provide a reference for the diagnostic criteria used for WD.
4. There is a reference to “Supplementary Materials,” but the Reviewer did not find it in the system.
5. Statistical analysis
a. What was the software (and edition) used for statistical analysis?
6. Discussion
a. It is advised to provide a section regarding the study’s limitations.
Round 2
Reviewer 1 Report
The authors did not address the comments and concerns
Author Response
Dear Reviewer:
We apologized that we didn’t give a satisfactory answer to your valuable comments. We think our paper should be had some improvement to meet the publishing requirements. Therefore, we have added some answers to your valuable suggestions and revised again our manuscript. In our opinion, the answers of the following valuable comments should have some improvements, and the point-by-point supplementary answers is proved as follow.
3- Line 101; Please justify the rational for use biceps brachii and the medialis and lateralis gastrocnemius muscle for biomechanical analysis.
Supplementary answer 3: Thanks again for your comments. In last time, we have explained that dystonia is one of the most common representations of Wilson’s disease (WD), and dystonia can appear in multiple parts of the body (including hand, upper extremity, face, leg and so on) in WD[1-3]. As a special neurogenetic disease, dystonia of extremities is also common in WD patients reported by previous studies, and its impact on the quality of life and prognosis of patients is particularly prominent[4-5], which are consistent with our clinical observation. And our previous study has found that the digital muscle function assessment system (MyotonPRO) can be used to evaluate the degree of upper limb dystonia and the efficacy of high frequency repetitive transcranial magnetic stimulation(rTMS) in the treatment of upper limb dystonia by measuring the bilateral biceps brachii in WD patients[6]. Therefore, we selected the extremities to measure the biomechanical level of WD patients, and the muscle biomechanical level was assessed using the MyotonPRO to reflect the degree of dystonia. Since this assessment system operates by giving a small mechanical impact to the tissue, the resulting oscillation of the tissues is recorded and the parameters are calculated from this curve. So, it requires that the measured muscle should be superficial and should not be too small and thin. Considering the above factors, we chose the biceps brachii and the medialis and lateralis gastrocnemius muscle on both sides for measurement to reflect the degree of the extremities dystonia. We have added some explanations in clinical and muscle biomechanical assessment part.
4- I strongly recommend the authors to report the study according to STROBE stamen.
Supplementary answer 4: Thank you very much for your valuable suggestion again. We revised our paper again which make our paper more meet the STROBE statement.
5- Is the sample size based on power calculations.
Supplementary answer 5: Thank you again for your comments. In last time, we have explained that it’s difficult to recruit patients for this study because Wilson’s disease is a rare disorder, and the results of clinical data and functional connectivity are statistically significant after by a strict corrected method. However, the sample size of clinical study should be based on power calculations. Here, the d value of the Cohen was calculated between two groups based on muscle parameters, and found the value of Cohen’d is 0.341. We sincerely know that we should expand the sample size to make our study meet standard of power calculation. Although the results of fMRI in present study is statistically significant, we should add the question of sample size in the limitation of present study. Thanks again.
7- Although the study is interesting and informative, the discussion is the weakest part. There is no good explanation of the results, clinical implication, or limitation.
Supplementary answer 7: Thanks again for your comments again. We have added some result explanations in discussion part. But we think this part should have some improvements, particularly in explanation of FC between lenticular nucleus and cerebellum. We have added some extra explanations of this results in discussion section.
In addition, we also have polished our English expression and revised some gramma mistakes in our manuscript. Thanks again for review our paper and your valuable comment help us a lot to improve the quality of the paper.
References
[1] Członkowska A, Litwin T. Wilson disease - currently used anticopper therapy.[J]. Handbook of clinical neurology, 2017, 142181-191.
[2] Członkowska A, Litwin T, Dzieżyc K, et al. Characteristics of a newly diagnosed Polish cohort of patients with neurological manifestations of Wilson disease evaluated with the Unified Wilson's Disease Rating Scale.[J]. BMC neurology, 2018, 18(1): 34.
[3] Członkowska A, Litwin T, Dusek P, et al. Wilson disease.[J]. Nature reviews. Disease primers, 2018, 4(1): 21.
[4] Lozeron P, Poujois A, Meppiel E, et al. Inhibitory rTMS applied on somatosensory cortex in Wilson's disease patients with hand dystonia.[J]. Journal of neural transmission (Vienna, Austria : 1996), 2017, 124(10): 1161-1170.
[5] Cao Z, Rao R, Wu T, et al. Botulinum toxin type A treatment of four cases of Wilson disease with lower limb dystonia: A prospective study.[J]. Toxicon : official journal of the International Society on Toxinology, 2022, 106959.
[6] Hao W, Wei T, Yang W, et al. Effects of High-Frequency Repetitive Transcranial Magnetic Stimulation on Upper Limb Dystonia in Patients With Wilson's Disease: A Randomized Controlled Trial.[J]. Frontiers in neurology, 2021, 12783365.

Reviewer 2 Report
The authors have addressed all the points that I have raised.
Author Response
I'm glad to be recognized by you. Thank you for your contribution and work in this paper. I wish you a happy life and smooth work!